# Methane formation driven by light and heat prior to the origin of life and beyond

Leonard Ernst [1,2] ✉, Uladzimir Barayeu [3,4], Jonas Hädeler [5], Tobias P. Dick[3,4], Judith M. Klatt[2,6,7], Frank Keppler [5,8] & Johannes G. Rebelein [1,2] ✉

Methane is a potent greenhouse gas, which likely enabled the evolution of life by keeping the early Earth warm. Here, we demonstrate routes towards abiotic methane and ethane formation under early-earth conditions from methylated sulfur and nitrogen compounds with prebiotic origin. These compounds are demethylated in Fenton reactions governed by ferrous iron and reactive oxygen species (ROS) produced by light and heat in aqueous environments. After the emergence of life, this phenomenon would have greatly intensified in the anoxic Archean by providing methylated sulfur and nitrogen substrates. This ROS-driven Fenton chemistry can occur delocalized from serpentinization across Earth's humid realm and thereby substantially differs from previously suggested methane formation routes that are spatially restricted. Here, we report that Fenton reactions driven by light and heat release methane and ethane and might have shaped the chemical evolution of the atmosphere prior to the origin of life and beyond.

Methane ($CH_4$) is a potent greenhouse gas which has in the past and is still today contributing to climate change[1]. Atmospherically accumulated $CH_4$ and ethane ($C_2H_6$) might also explain the "faint young sun paradox", which describes the apparent contradiction of a fainter sun (70 – 83% of the current solar energy output) but a climate that was at least as warm as today during early Earth (4.5–2.5 Ga ago)[2–4]. Although these $CH_4$ levels would be essential to keep the Earth a liquid hydrosphere to allow the evolution of life during the Archean (4.0–2.5 Ga), the source of $CH_4$ prior to the origin of life is still under debate[5]. While $CH_4$ was released by submarine volcanism, most $CH_4$ is suggested to be formed as side product of serpentinization[5]. After the evolution of microbial methanogenesis latest by 3.5 Ga[6], methanogenesis could have been responsible for a $CH_4$ flux comparable to today[7]. Thus, methanogenesis is expected to be the main source of $CH_4$ during the Archean, supported by light carbon isotope values in sedimentary deposits[8]. However, isotope signals can only manifest upon reoxidation and $CH_4$ itself does not leave much of a signature in the geological record. Thus, the actual $CH_4$ concentrations and the potential abiotic sources during early Earth remain elusive. Based on mass-independent fractionation of sulfur, at least 20 ppmv $CH_4$ was present around 2.4 Ga ago[9]. A more recent study analyzing the fractionation of xenon isotopes suggests $CH_4$ levels of >5000 ppmv around 3.5 Ga ago[10]. Catling et al. expect even higher $CH_4$ levels at the beginning of the Archean (4 Ga)[3] before methanogenesis evolved. Yet, the processes responsible for these high $CH_4$ levels and their relative contributions remain controversial.

Recently, we discovered a non-enzymatic $CH_4$ formation mechanism expected to occur in all living organisms[11]. The mechanism has been demonstrated to be active in over 30 very diverse organisms[11] and suggested to explain previously observed $CH_4$ formation by cyanobacteria[12], freshwater and marine algae[13,14], saprotrophic fungi[15] and plants[16]. The $CH_4$ formation is driven by a cascade of radical reactions, governed by the interplay of reactive oxygen species (ROS) and ferrous iron ($Fe^{2+}$), methylated sulfur (S)- and nitrogen (N)-compounds

[1]Max Planck Institute for Terrestrial Microbiology, 35043 Marburg, Germany. [2]Center for Synthetic Microbiology (SYNMIKRO), 35032 Marburg, Germany. [3]Division of Redox Regulation, German Cancer Research Center (DKFZ), DKFZ-ZMBH Alliance, 69120 Heidelberg, Germany. [4]Faculty of Biosciences, Heidelberg University, 69120 Heidelberg, Germany. [5]Institute of Earth Sciences, Heidelberg University, 69120 Heidelberg, Germany. [6]Microcosm Earth Center, Max Planck Institute for Terrestrial Microbiology & Philipps University Marburg, 35032 Marburg, Germany. [7]Biogeochemistry Group, Department for Chemistry, Philipps University Marburg, 35032 Marburg, Germany. [8]Heidelberg Center for the Environment HCE, Heidelberg University, 69120 Heidelberg, Germany. ✉e-mail: leonard.ernst@mpi-marburg.mpg.de; johannes.rebelein@mpi-marburg.mpg.de

are oxidatively demethylated by hydroxyl radicals ($\cdot$OH) and oxo-iron(IV) complexes ($[Fe^{IV}=O]^{2+}$) to yield methyl radicals ($\cdot CH_3$)[11].

Here we show that this abiotic mechanism occurs also outside living cells and might have contributed to $CH_4$ levels before life emerged. All needed components: (i) methylated S- and N-compounds, (ii) $Fe^{2+}$ and (iii) ROS are found under early-earth conditions. (i) In a prebiotic world, methylated S-compounds like methanethiol, dimethyl sulfide (DMS) or dimethyl sulfoxide (DMSO) were formed abiotically under the reducing conditions of hydrothermal vents[17–19] or transported to Earth by carbonaceous meteorites during early Earth meteorite bombardment[20,21]. Upon the emergence of life, more methylated S-/N-compounds were produced by cells and organisms, i.e. methionine, dimethylsulfoniopropionate or trimethylamine[22]. (ii) Under the anoxic conditions of the early Earth, oceans were rather ferruginous, i.e. rich in $Fe^{2+}$ required for Fenton chemistry[23,24], nonetheless ferric iron ($Fe^{3+}$) also occurred in Archean seawater[25]. Additionally, the mechanism driven by $Fe^{2+}$ can be enhanced by Fenton-promoting $Fe^{2+}$-chelators, e.g. ATP or citrate[26]. Under anoxic conditions, Fe(III)-carboxylate complexes are photochemically reduced via ligand-to-metal charge transfer (LMCT)[27], resulting in $Fe^{2+}$ and organic radicals[28]. (iii) Under ambient temperatures, low ROS levels exist in water that increase with heat[29], or can be generated by photolysis or radiolysis[30–33]. Under acidic

conditions, i.e. in volcanic lakes[34], illumination of Fe(III)-aqua complexes ($[Fe(H_2O)_6]^{3+}$) forms $Fe^{2+}$ and ROS[35,36]. Thus, we hypothesized that the Fenton reaction of $Fe^{2+}$ with $H_2O_2$, generated by heat and light, could have driven the formation of $CH_4$ from methylated S-/N-compounds independent of temperatures and pressures occurring at hydrothermal vents but at ambient conditions as early as the prebiotic world of the Hadean (4.5–4.0 Ga, Fig. 1a). To identify critical components of such a mechanism, we used aqueous model systems to determine the influence of heat, light, and (bio)molecules on $CH_4$ formation in abiotic and biotic environments.

## Results

### Methane is formed under abiotic conditions

To investigate $CH_4$ formation under abiotic conditions (Fig. 1a), we designed a chemical model system consisting of a nitrogen atmosphere, a potassium phosphate-buffered solution (pH 7, expected during the Archean at 4.0 Ga[37]) supplemented with $Fe^{2+}$ and the abiotically formed DMSO which serves as methyl donor for ROS-driven $CH_4$ formation. Over the course of the experiments, no pH change was observed, while low amounts of Fe(OH)$_2$ precipitated. In this model system, $CH_4$ was consistently formed from DMSO in the dark (Fig. 1b). $CH_4$ formation rates increased with rising temperatures from 30 to 97 °C, consistent

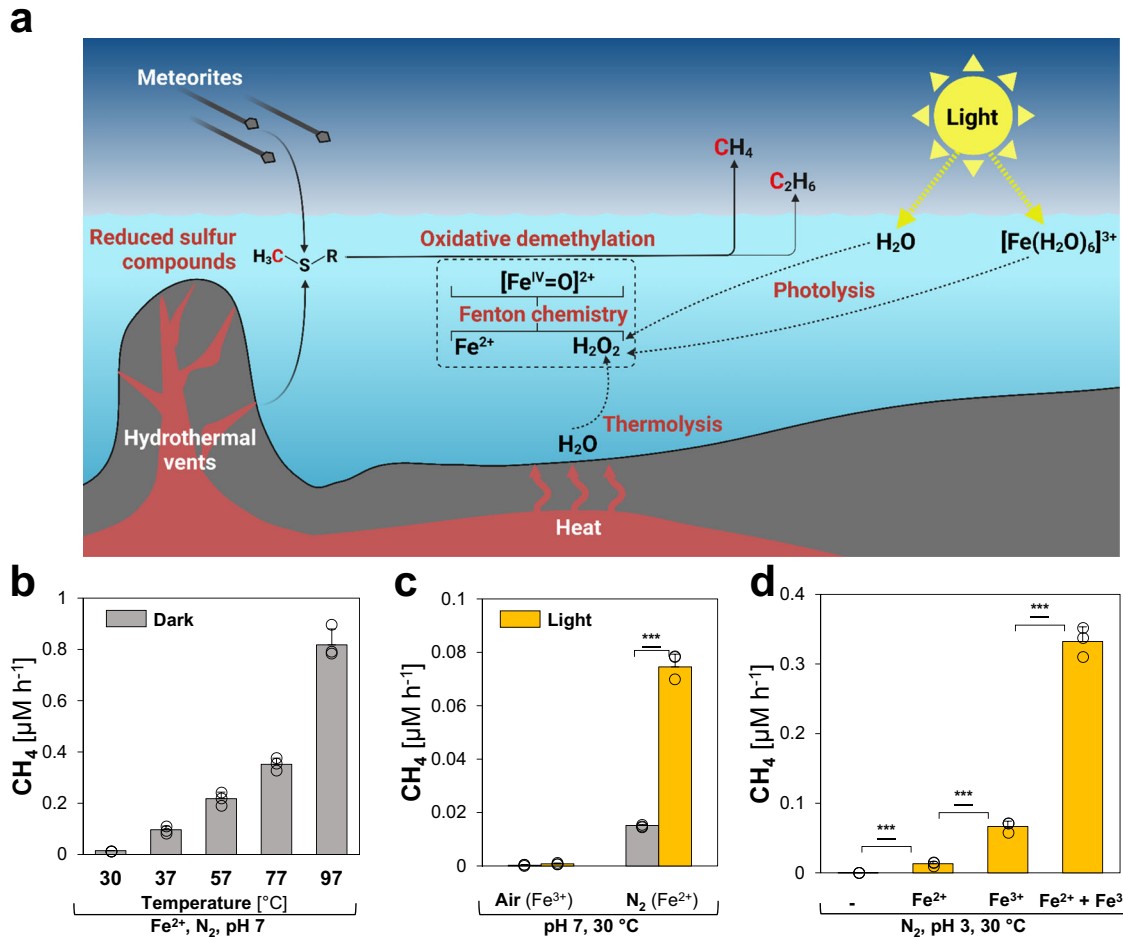

**Fig. 1 | Heat and light drive $CH_4$ formation under abiotic conditions. a** Reduced, methylated S-/N-compounds are formed abiotically in hydrothermal vents or transported to Earth by carbonaceous meteorites. Under anoxic conditions, $H_2O_2$ is formed by thermolysis and photolysis of water and $[Fe(H_2O)_6]^{3+}$ complexes, reacting with dissolved ferrous iron ($Fe^{2+}$) to hydroxyl radicals ($\cdot$OH) and $[Fe^{IV}=O]^{2+}$ compounds that drive the oxidative demethylation of methylated S-/N-compounds, thereby facilitating $CH_4$ and $C_2H_6$ formation. **b** Thermolysis: $CH_4$ is formed from DMSO under high temperatures. **c** Water photolysis: The formation of $CH_4$ is

increased by light. **d** $[Fe(H_2O)_6]^{3+}$ photolysis: Under acidic conditions, light-driven $CH_4$ formation is enhanced by $[Fe(H_2O)_6]^{3+}$ photochemistry. All experiments were conducted in closed glass vials containing buffered solutions (pH 7 or pH 3) supplemented with DMSO and $Fe^{2+}$ or $Fe^{3+}$ at 30 °C (**b**, **c**) under a $N_2$ or air atmosphere. Statistical analysis was performed using paired two-tailed $t$ tests, ***$p \le 0.001$. The bars are the mean + standard deviation of triplicates, shown as circles. **a** Was created with BioRender.com.

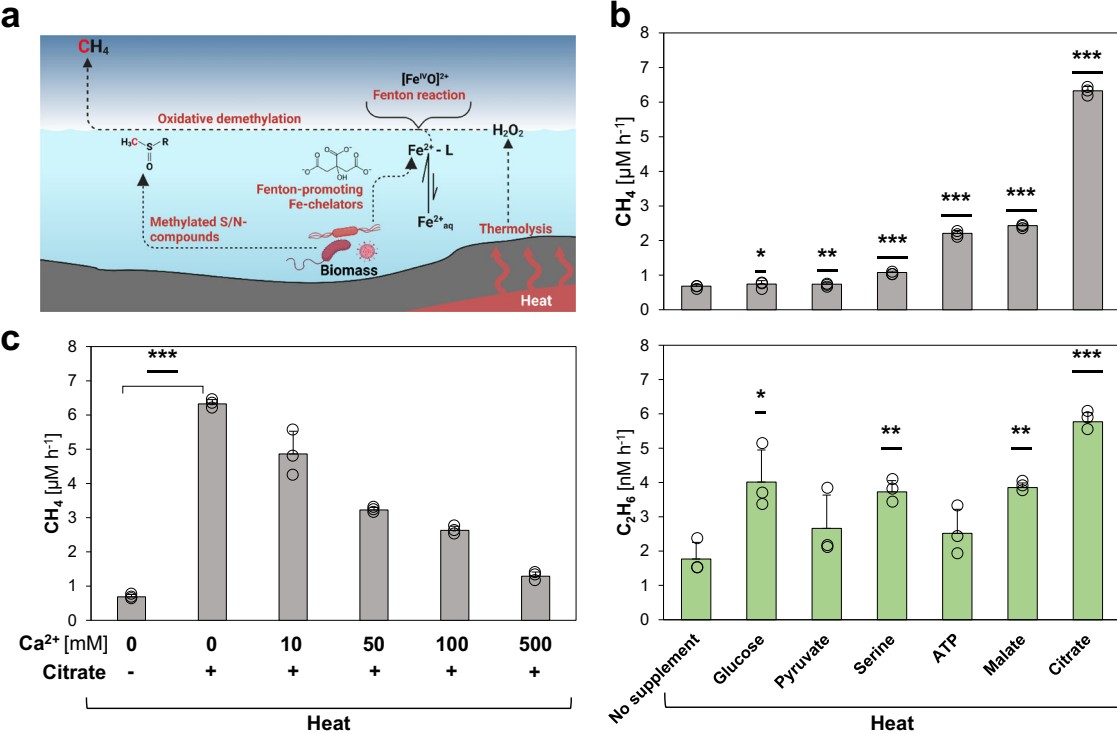

**Fig. 2 | (Bio)molecules enhance heat-driven CH$_4$ formation. a** Overview of CH$_4$ formation driven by heat. Living organisms produce S-/N-methylated compounds that serve as substrates for CH$_4$ formation and Fe$^{2+}$-chelators that promote Fenton chemistry and enhance CH$_4$ formation. **b** Heat-driven CH$_4$ (upper panel) and C$_2$H$_6$ (lower panel) formation is enhanced upon supplementation with (bio)molecules. **c** Citrate enhances heat-driven CH$_4$ formation acting as iron-chelator. Upon the addition of Ca$^{2+}$, CH$_4$ levels decrease due to the replacement of Fenton-promoting Fe$^{2+}$-citrate complexes with Ca$^{2+}$-citrate complexes. All experiments were conducted in closed glass vials containing a buffered solution (pH 7) supplemented with DMSO, Fe$^{2+}$ and, optionally, citrate and Ca$^{2+}$ under a pure nitrogen atmosphere at 97 °C (heat). Statistical analysis was performed using paired two-tailed *t* tests, \**p* ≤ 0.05, \*\**p* ≤ 0.01, \*\*\**p* ≤ 0.001. The bars are the mean + standard deviation of triplicates, shown as circles. **a** Was created with BioRender.com.

with the previously reported temperature-dependency of ROS levels in water[29]. While only marginal CH$_4$ formation rates derived from DMSO were observed at 30 °C (~0.02 μM h$^{-1}$), rates increased 41-fold to ~0.82 μM h$^{-1}$ at 97 °C. In addition, low C$_2$H$_6$ amounts were formed (Supplementary Fig. 1), most likely resulting from the recombination of two methyl radicals[23]. At 37 °C, the CH$_4$:C$_2$H$_6$ ratio was ~110, with an increasing trend towards higher temperatures. As the ROS-driven CH$_4$:C$_2$H$_6$ ratios are substantially lower than those observed for archaeal methanogenesis[38], the CH$_4$:C$_2$H$_6$ ratios could serve as indicator to distinguish microbial from abiotic processes.

Light enhanced the abiotic CH$_4$ formation rates (Fig. 1c) by photolysis of water and generation of H$_2$O$_2$ at 30 °C (Supplementary Fig. 2). Notably, CH$_4$ amounts increased ~4-fold from ~0.02 μM h$^{-1}$ to ~0.08 μM h$^{-1}$ upon broad-spectrum illumination (~350 nm < λ < ~1010 nm at 82 ± 4 μmol photons m$^{-2}$s$^{-1}$, Supplementary Fig. 3). This data provides evidence that light-driven CH$_4$ formation from methylated S-compounds can occur even in the absence of biomolecules. The addition of oxygen to the samples stopped the formation of CH$_4$ in this pH-neutral model system supplemented with Fe$^{3+}$ (Fig. 1c). In contrast, under acidic (pH 3), illuminated conditions CH$_4$ formation rates increased ~5-fold upon Fe$^{3+}$-supplementation in comparison to Fe$^{2+}$-addition, indicating light-driven ROS and Fe$^{2+}$ formation from [Fe(H$_2$O)$_6$]$^{3+}$ complexes (Fig. 1d)[35]. Upon supplementation of 1 mM Fe$^{3+}$ and 1 mM Fe$^{2+}$, keeping the overall iron concentration unchanged at 2 mM, CH$_4$ formation rates increased to ~0.33 μM h$^{-1}$. This 5-fold rate increase is driven by both ROS-inducing Fe$^{3+}$ and Fenton-driving Fe$^{2+}$. Under pH-neutral conditions, mixing Fe$^{2+}$ and Fe$^{3+}$ only increased CH$_4$ formation rates by ~1.3-fold in comparison to Fe$^{2+}$-supplemented samples, while only trace amounts of CH$_4$ were obtained from Fe$^{3+}$-supplemented samples (Supplementary Fig. 4).

Thus, illuminated [Fe(H$_2$O)$_6$]$^{3+}$ complexes generate both Fe$^{2+}$ and ROS, thereby contributing to the ROS-driven CH$_4$ formation under acidic conditions.

Taken together, we demonstrated that heat and light drive the formation of CH$_4$ and C$_2$H$_6$ in an anoxic, abiotic environment under ambient temperatures and pressures. These results establish a ROS-driven mechanism based on Fenton chemistry that can occur delocalized from serpentinization across Earth's humid realm and thereby substantially differs from previously suggested mechanisms that are spatially restricted. Thus, this non-enzymatic hydrocarbon formation mechanism could have released CH$_4$ and C$_2$H$_6$ into the atmosphere of the Hadean and Archean. Besides CH$_4$, C$_2$H$_6$ is considered an important factor in keeping the early Earth warm, since C$_2$H$_6$ absorbs from 11 to 13 μm in an atmospheric window (roughly 8–13 μm) where H$_2$O and CO$_2$ do not absorb strongly[2]. Together, the hydrocarbons produced by these pathways might offer a solution to the "faint young sun paradox"[3,4].

## (Bio)molecules enhance the heat-driven CH$_4$ formation

Even before life emerged, several metabolites, e.g. citrate and malate, could have been formed via an ancient, non-enzymatic TCA cycle predecessor driven by ROS[39,40]. Catalyzed by iron particles, the formation of pyruvate from CO$_2$ was recently reported[41]. Intriguingly, citrate and malate, as well as other primordial (bio)molecules with a putative prebiotic origin, including ATP[42] or serine[43], have been reported to act as Fenton-promoting Fe$^{2+}$-chelators[26]. We therefore investigated if these hydroxylated and carboxylated (bio)molecules enhance the ROS-driven CH$_4$ formation rates (Fig. 2a).

Indeed, the addition of pyruvate, glucose, serine, ATP, malate or citrate to the heat-driven (97 °C) model system increased the abiotic CH$_4$

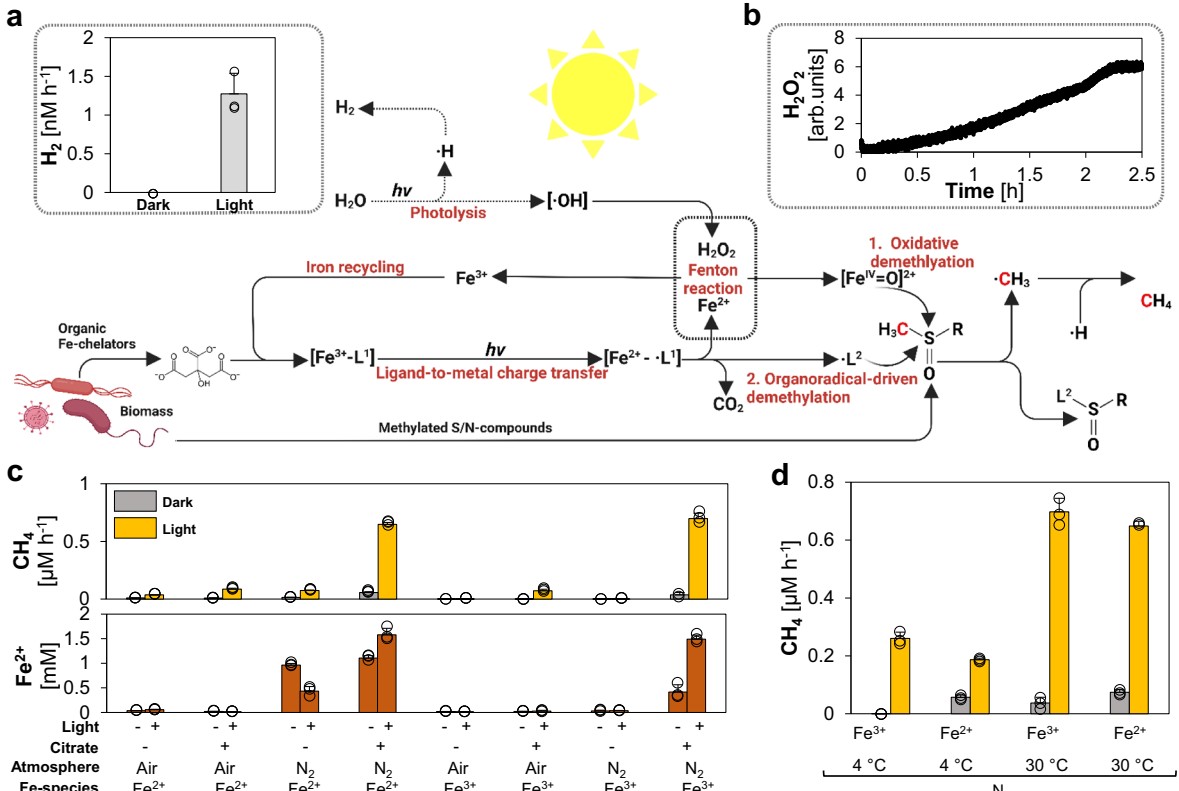

**Fig. 3 | A light-driven iron redox cycle drives and enhances CH₄ formation.**
Upon illumination, water is photolytically split into hydroxyl radicals (·OH) and hydrogen forming $H_2$ and $H_2O_2$. Organic $Fe^{3+}$-complexes ($Fe^{3+}$-[$L^1$]) are converted into $Fe^{2+}$ and organic radicals (·$L^1$) via ligand-to-metal charge transfer (LMCT). The generated $Fe^{2+}$ reacts with $H_2O_2$ to ·OH or [$Fe^{IV}$= O]$^{2+}$ and thereby drives the generation of methyl radicals (·$CH_3$) from S-/N-methylated compounds. The LMCT-generated ·$L^1$ decomposes into $CO_2$ and another organic radical (·$L^2$) that additionally facilitates CH₄ formation upon reacting with S-/N-methylated compounds. Under light, (**a**) H₂ (gray bars) and (**b**) $H_2O_2$ is formed in pure buffer. **c** Upon

illumination, CH₄ formation rates (yellow bars) are increased. $Fe^{2+}$ formation (brown bars) depends on anoxic conditions and is driven by LMCT induced by the addition of citrate. **d** Light and heat have synergistic effects on CH₄ formation. While heat drives CH₄ formation upon $Fe^{2+}$-supplementation, light increases CH₄ formation upon $Fe^{3+}$- and $Fe^{2+}$-addition. All experiments were conducted in closed glass vials containing a buffered solution (pH 7) supplemented with DMSO, $Fe^{3+}$ or $Fe^{2+}$, $N_2$ or air atmosphere in the presence or absence of citrate incubated under light or in the dark at 4 °C or 30 °C. The bars are the mean + standard deviation of triplicates, shown as circles. **a**, **b** Was created with BioRender.com.

formation rate, e.g. more than 11-fold for citrate (Fig. 2b). Corresponding $C_2H_6$ rates significantly increased for glucose, serine, malate and citrate, resulting in CH₄:$C_2H_6$ ratios between ~190 (glucose) and ~1100 (citrate, Fig. 2b). To test if these enhancing effects were indeed driven by $Fe^{2+}$ chelation, we supplemented the assays with the $Fe^{2+}$-competitor $Ca^{2+}$ (Fig. 2c). Since (bio)molecules like citrate can alternatively chelate $Ca^{2+}$ ions, we expected that increasing $Ca^{2+}$ concentrations result in decreasing CH₄ formation rates by replacing Fenton-promoting $Fe^{2+}$-citrate complexes with $Ca^{2+}$-citrate complexes. Upon addition of 10 mM and 500 mM $Ca^{2+}$, CH₄ formation rates significantly decreased from ~6.32 μM h⁻¹ to ~4.86 μM h⁻¹ and ~1.29 μM h⁻¹, respectively. Thus, 500 mM $Ca^{2+}$ suppressed ~90% of the Fenton-promoting effect of citrate supplementation. The $Ca^{2+}$ concentration-dependent decrease of the heat-driven CH₄ formation rate supports the role of citrate as a Fenton-promoting $Fe^{2+}$-chelator, which is further indicated by citrate dissolving any ferruginous precipitate.

Together, ROS generated by heat interact with iron and thereby drive the formation of methyl radicals from S-/N-methylated compounds, resulting in CH₄ and $C_2H_6$. Moreover, several hydroxylated or carboxylated (bio)molecules with a putative prebiotic origin were shown to act as Fenton-promoting $Fe^{2+}$-chelators, indicating that ROS-driven CH₄ formation may have already been widespread within the timeframe of the transition from prebiotic chemistry to the origin of life. The rise of life would have fostered the abiotic, non-enzymatic CH₄ formation due to the consequential formation and release of biomolecules serving as chelators and substrates.

## A light-driven iron redox cycle sustains CH₄ formation
During Fenton chemistry, $Fe^{2+}$ is either oxidized to [$Fe^{IV}$= O]$^{2+}$ or ferric iron ($Fe^{3+}$). As $Fe^{3+}$ cannot drive Fenton reactions[23,24], CH₄ formation rates decrease with increasing reaction time and increasing concentrations of $Fe^{3+}$. While this effect may have been minor in the ferruginous Archean oceans, $Fe^{3+}$ likely dominated the iron pool in the photic zone of the oceans latest by the rise of photoferrotrophy and was also prevalent in several ecological niches, e.g. volcanic lakes[34]. The evolution of photosynthesis and the subsequent biological production of $O_2$ oxidized the majority of the available $Fe^{2+}$ to $Fe^{3+}$. Thus, abiotic ROS-driven CH₄ formation would have been hindered in the sunlit realm by the late Archean in the absence of an iron redox cycle at neutral pH. Intriguingly, besides acting as Fenton-promoting $Fe^{2+}$-chelators[26], (bio)molecules like citrate were reported to reduce $Fe^{3+}$ to $Fe^{2+}$ via LMCT under oxic and anoxic conditions[27]. Therefore, (bio)molecules may have facilitated widespread iron redox cycling, e.g. by forming Fe(III)-carboxylate complexes. Furthermore, previous studies showed that, upon illumination of water hydroxyl radicals (·OH) and hydrogen atoms are generated, forming $H_2O_2$ and $H_2$[30–33]. Thus, we hypothesized that light could drive CH₄ formation in the absence of $Fe^{2+}$ by simultaneously (i) generating ROS from water and (ii) reducing $Fe^{3+}$ to $Fe^{2+}$ via LMCT, thereby recycling $Fe^{3+}$ and keeping the Fenton reaction running (Fig. 3).

To verify our hypothesis, we first confirmed light-dependent ROS production in our model system in the absence of substrate, iron and organic ligands by measuring final reaction products of

photolysis: $H_2$ and $H_2O_2$ (Fig. 3a, b). We measured $H_2$ production at a rate of ~1.3 nM $h^{-1}$ in anoxic samples under broad-spectrum illumination but not in samples kept in the dark (Fig. 3a). A continuous formation of $H_2O_2$ was measured online using microsensors, which confirmed light-dependent production dynamics in pure buffer (Fig. 3b). Via fluorescence-based $H_2O_2$ endpoint measurements, we found that both iron and DMSO reduced the $H_2O_2$ concentrations. The decrease in $H_2O_2$ levels can be attributed to Fenton reactions between $H_2O_2$, $Fe^{2+}$ and the radical scavenger DMSO (Supplementary Fig. 5).

Building on this, we closely investigated the interplay of LMCT and iron photochemistry on $CH_4$ formation. For this purpose, we analyzed our chemical model system containing a buffered solution (pH 7), $Fe^{2+}$ or $Fe^{3+}$, DMSO, in the presence or absence of citrate for the formation of $CH_4$ and the concentration of available $Fe^{2+}$ (Fig. 3c). The influence of the following parameters on the formation of $CH_4$ was tested: (i) $O_2$ ( ~ 21% in air), (ii) oxidation state of the supplemented iron species ($Fe^{2+}$ *vs.* $Fe^3$), (iii) light and (iv) presence/absence of citrate. (i) $CH_4$ formation rates under anoxic conditions always exceeded rates under oxic conditions. (ii) Without citrate, initial $Fe^{2+}$-supplementation was required to form significant $CH_4$ levels. (iii) $CH_4$ formation always increased with light. (iv) Upon citrate addition, $CH_4$ formation was enhanced in illuminated and anoxic samples containing DMSO and $Fe^{2+}$ or $Fe^{3+}$. Besides elevated $CH_4$ formation rates, citrate addition also increased the final $Fe^{2+}$ concentrations, e.g. from ~0 mM $Fe^{2+}$ to ~1.5 mM $Fe^{2+}$ in illuminated and anoxic samples.

After determining the influence of the four parameters (i) $O_2$, (ii) iron (iii) light and (iv) (bio)molecules, we further investigated them individually to gain a better understanding of their contribution and role in the light-driven $CH_4$ formation.

(i) $O_2$: The influence of $O_2$ on LMCT and $CH_4$ formation was studied in citrate-supplemented samples by adding various amounts of air. $Fe^{2+}$ concentrations and $CH_4$ formation rates decreased with increasing $O_2$ levels (Supplementary Fig. 6). In comparison to 0 % $O_2$, the $Fe^{2+}$ concentration dropped drastically already at 0.2 % $O_2$ and was ~96 % lower at 2 % $O_2$, while $CH_4$ formation rates decreased approximately linearly with the $O_2$ level. This indicates the presence of a Fe-cycle, in which most LMCT-formed $Fe^{2+}$ is instantly re-oxidized, either by $O_2$ or Fenton reactions. The balance between these $Fe^{2+}$ sinks depend on $O_2$ availability and governs $CH_4$ formation rates. In the presence of $O_2$, we also detected methanol ($CH_3OH$) formation rates ranging from ~0.003 µM $h^{-1}$ (0.2 % $O_2$) to ~0.07 µM $h^{-1}$ (21 % $O_2$). $CH_3OH$ is preferentially formed through the reaction of ·$CH_3$ with $O_2$[23,44]. Without the addition of $O_2$, no $CH_3OH$ was detected, indicating anoxic conditions in our standard assays.

(ii) Iron: The role of the LMCT-rate and the corresponding $Fe^{2+}$ availability for $CH_4$ formation was tested by supplementing the assays with various $Fe^{3+}$ concentrations (Supplementary Fig. 7). At lower $Fe^{3+}$ concentrations, $CH_4$ formation rates increased steeper than the measured $Fe^{2+}$ concentrations. At high $Fe^{3+}$ concentrations, $CH_4$ formation rates leveled off, while $Fe^{2+}$ concentrations continued to increase. This indicates that $Fe^{2+}$ is limiting the demethylation rates at low iron concentrations, because it is immediately re-oxidized, while light-dependent ROS production is limiting $CH_4$ formation at high iron concentrations. Most importantly, these data highlight that a light- and ROS-driven iron cycle can facilitate high rates of $CH_4$ formation, even in the presence of $O_2$ and the absence of detectable $Fe^{2+}$, which opens the possibility of widespread abiotic $CH_4$ production after the great oxidation event as well as in diverse modern habitats. Next, we investigated the role of the alkali metal magnesium ($Mg^{2+}$) due to its high environmental abundance and found that $Mg^{2+}$ does not facilitate $CH_4$ formation in illuminated buffer containing DMSO and citrate (Supplementary Fig. 8). Upon additional $Fe^{3+}$ supplementation, $Mg^{2+}$ also decreased $CH_4$ formation rates by replacing Fenton-promoting

$Fe^{3+}$-citrate complexes by $Mg^{2+}$-citrate complexes, thereby acting similar to $Ca^{2+}$ that was demonstrated to decrease heat-driven $CH_4$ formation (Fig. 2c). Besides iron, the transition metals copper, cerium, cobalt, nickel and manganese were reported to drive Fenton chemistry[45,46], resulting in the release of $CH_4$. Thus, we tested different transition metals in our chemical model system, containing DMSO as substrate and ascorbate as a strong metal reductant[47,48]. We observed that copper, cobalt and cerium also enhanced $CH_4$ formation rates (Fig. 4a). However, the activity of copper, cobalt and cerium was lower than iron. The high activity of iron combined with its ubiquitous abundance in the Precambrian highlights the global distribution and importance of this mechanism.

(iii) Light: It is established that light quality has an important influence on photolysis. Short wavelength light in the ultraviolet spectrum was reported to drive water photolysis and LMCT more efficiently than longer wavelengths[49]. We expected that shorter wavelength light would increase both $CH_4$ formation rates and $Fe^{2+}$ levels. Indeed, $CH_4$ formation rates surged from ~0.3 µM $h^{-1}$ ($\lambda_{max} = 534$ nm) to ~1.23 µM $h^{-1}$ ($\lambda_{max} = 388$ nm, Fig. 4b) and $Fe^{2+}$ concentrations almost tripled from ~1.3 mM ($\lambda_{max} = 534$ nm) to ~4.2 mM ($\lambda_{max} = 388$ nm). Although the broad-spectrum light had a 1.5-fold higher energy flux ($57 \pm 2$ kJ $m^{-2}$ $h^{-1}$) compared to the 388 nm-LED light ($37 \pm 2$ kJ $m^{-2}$ $h^{-1}$), the $CH_4$ formation rate under the broad-spectrum light was only half (0.7 µM $h^{-1}$). Given that the stratospheric ozone layer was absent during the Hadean and Archaean, higher fluxes of short wavelength light (*i.e.* ultraviolet light), reached aqueous environments and may have further enhanced the ROS-driven $CH_4$ formation.

(iv) (Bio)molecules: After illumination of $Fe^{3+}$-ligand complexes, one electron is transferred via LMCT from a carboxylated ligand ($L^1$) to $Fe^{3+}$, an organic radical (·$L^1$), i.e. citrate radical, is generated. As described in the literature[28], we observed the subsequent $CO_2$ disassembly from citrate radicals (Supplementary Fig. 9). We speculated that the remaining organic radical (·$L^2$) could react with DMSO, resulting in ·$CH_3$ and the formation of $CH_4$ (Fig. 3). Since we cannot directly detect organic radicals, we mimicked the proposed reaction in an anoxic model system only containing DMSO and the radical-generating 2,2'-azobis(2-amidinopropane) dihydrochloride (APPH) that readily decomposes into carbon-centered organic radicals at 40 °C (Supplementary Fig. 10). Indeed, we observed $CH_4$ formation in a mixture of DMSO and APPH, while only trace amounts of $CH_4$ were observed from either DMSO or AAPH alone, suggesting an organic radical-driven $CH_4$ formation mechanism. In short, carboxylates like citrate facilitate LMCT, thereby reducing $Fe^{3+}$ to $Fe^{2+}$ and forming organic radicals. Both resulting compounds drive $CH_4$ formation. Overall, $CH_4$ can be formed under anoxic conditions via (i) water thermolysis, (ii) water photolysis, (iii) $[Fe(H_2O)_6]^{3+}$ photolysis and (iv) LMCT-induced carbon-centered radicals. Apart from serving as chelators, some (bio)molecules could also serve as substrates for Fenton reactions. Thus, we investigated four S-/N-methylated compounds in the presence of the chelator citrate. Upon illumination, $CH_4$ was formed from dimethyl sulfide, methionine, 2-methylthioethanol and trimethylamine (Fig. 4c). These observations indicate that ROS-driven $CH_4$ formation significantly increased after the origin of life by providing biomolecules as chelators and substrates.

Finally, synergistic effects between light and heat were observed (Fig. 3d). For $Fe^{2+}$-supplemented samples, $CH_4$ rates at 4 °C increased from ~0.056 µM $h^{-1}$ in the dark over ~0.19 µM $h^{-1}$ under light to ~0.65 µM $h^{-1}$ in illuminated samples at 30 °C. For $Fe^{3+}$-supplemented samples, only $CH_4$ rates below 0.03 µM $h^{-1}$ were obtained in the dark, while $CH_4$ formation rates were slightly above $Fe^{2+}$-supplemented samples in the light, again demonstrating the effects of LMCT and LMCT-induced carbon-centered radicals. Thus, the two factors heat and light synergistically combine for a stable and enhanced ROS and $CH_4$ formation.

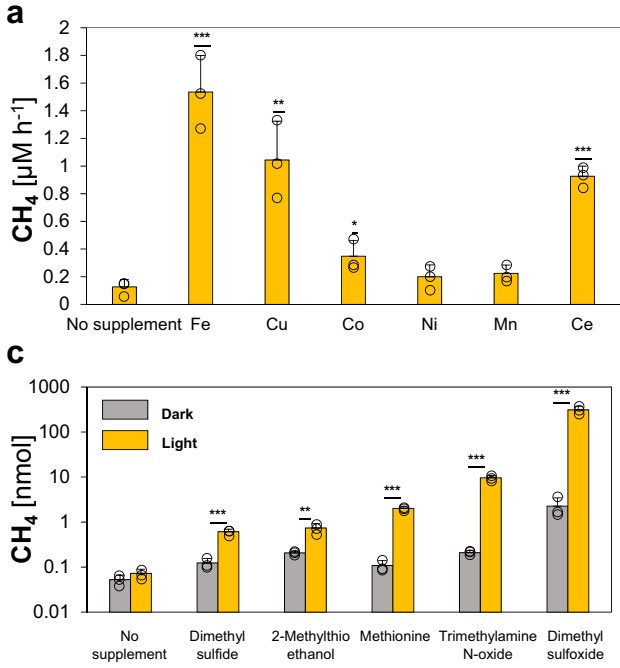

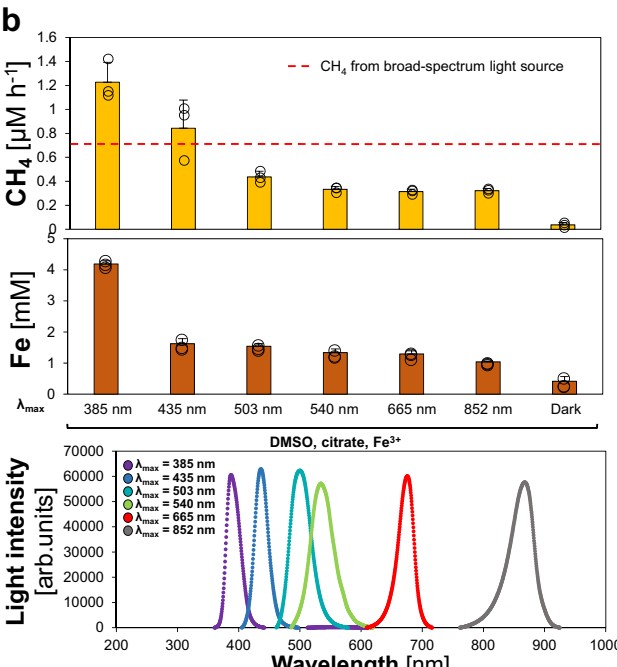

**Fig. 4 | Transition metals, wavelengths and methylated sulfur- and nitrogen compounds mediate light-driven CH₄ formation. a** Iron, cobalt and cerium enhance light-driven CH₄ formation. No significant CH₄ increase was observed for cobalt, nickel and manganese supplementation. **b** Light-driven CH₄ formation and $Fe^{2+}$ generation increases in the near-UV spectrum. **c** Light-driven formation of CH₄ from methylated S-/N-compounds (logarithmic scale). Upon illumination, significant increases in CH₄ levels were measured for dimethyl sulfide, methionine, 2-methylthioethanol, trimethylamine N-oxide and dimethyl sulfoxide (DMSO). All experiments were conducted in closed glass vials containing a buffered solution (pH 7), N₂ and either $Fe^{3+}$ or other transition metals (**a**), DMSO or other substrates (**c**) and either ascorbate (**a**) or citrate (**b**, **c**). Samples were incubated under broad-spectrum light (**a**, **c**), specific wavelengths (**b**) or in the dark at 30 °C. The dashed red line depicts the average CH₄ amounts obtained from samples illuminated by a broad-spectrum light source. Statistical analysis was performed using paired two-tailed *t* tests, *$p \leq 0.05$, **$p \leq 0.01$, ***$p \leq 0.001$. The bars are the mean + standard deviation of triplicates, shown as circles.

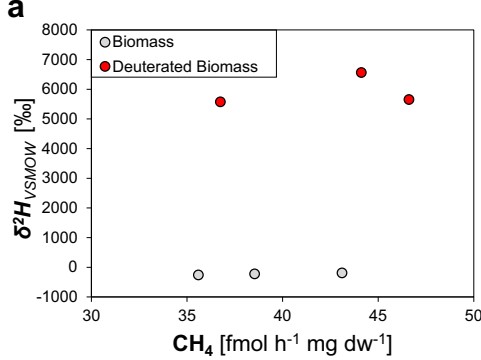

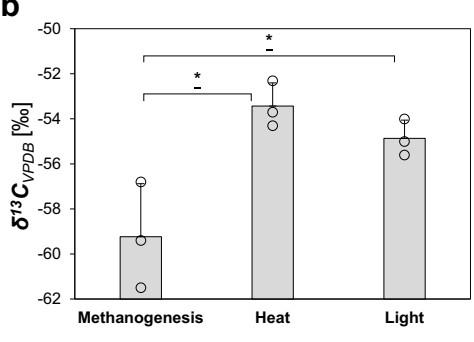

**Fig. 5 | Isotope labeling studies confirm dead biomass as substrate and show an abiotic isotope fractionation for ROS-driven CH₄ formation. a** Unlabeled or deuterium-enriched CH₄ is formed from unlabeled biomass (gray dots) or deuterated biomass (red dots), respectively. **b** Stable carbon isotope values of cultures from the methanogen *Methanothermobacter marburgensis*, heat-, or light-generated CH₄. All experiments were conducted in closed glass vials containing a buffered solution (**a**, **b**—heat, light) or culture medium (**b**—methanogenesis), supplemented with $Fe^{3+}$ and ascorbate (**a**) or $Fe^{2+}$ and citrate (**b**—heat, light) under a nitrogen atmosphere, incubated under light at 30 °C or in the dark at 97 °C. Statistical analysis was performed using paired two-tailed *t* tests, *$p \leq 0.05$. The bars are the mean + standard deviation of triplicates, shown as circles.

## Biomass-derived CH₄ with an abiotic isotope fractionation

Considering the impact of (bio)molecules on the LMCT-driven Fenton reaction, organic radical generation and the role of biomolecules as substrates, we expect the discussed mechanisms to have played and still play the most important role in the vicinity of decaying biomass. To demonstrate that CH₄ is indeed formed from dead biomass in the presence of a variety of biomolecules and not just in our well-defined model systems, we conducted deuterium labeling experiments. For this purpose, we grew the bacterium *B. subtilis* in *Luria-Bertani* medium supplemented with 10% D₂O and inactivated the cells by sonication and freezing (see Methods).

The obtained dead biomass was supplemented with $Fe^{3+}$ and ascorbate and incubated under broad-spectrum light. Around 40 fmol CH₄ h⁻¹ mg⁻¹ dry weight was obtained from labeled and unlabeled biomass (Fig. 5a). In addition, stable hydrogen isotope values ($\delta^2H$) of CH₄ from D₂O-treated biomass showed strong enrichment in deuterium (~5900 ‰) in comparison to unlabeled biomass (~−225 ‰), demonstrating a direct conversion of isotopically labeled biomass to

$CH_4$. This suggests that the availability of biomass, upon the emergence of life, has increased the $CH_4$ formation by delivering both (i) S-/N-methylated compounds and (ii) Fenton-promoting iron chelators. The presence of $CH_4$ has been suggested to be crucial for the evolution of life, since it could serve as life´s first carbon source via methanotrophy[50–52]. Following this line of thought, we could demonstrate that methanotrophic *Methylocystis hirsuta* grew on $CH_4$ generated by our light-driven model system, transferred to the headspace of the *M. hirsuta* culture (Supplementary Fig. 11). In fact, the "last methane-metabolizing ancestor" had likely the genes to perform methanogenesis and anaerobic methane oxidation[53], suggesting that, under high $CH_4$ concentrations, methanotrophy could have emerged prior to methanogenesis.

Finally, we speculated that ROS-driven $CH_4$ formation leads to different stable carbon isotope values ($\delta^{13}C$) compared to biological processes, *i.e.*, methanogenesis. The observed $\delta^{13}C$ values for $CH_4$ generated by heat or light were less negative (~$-54 \pm 1.1$‰) compared to the $\delta^{13}C$ value of the methanogen *Methanothermobacter marburgensis* (~$-59.2 \pm 2.3$‰, Fig. 5b). While the isotopic fractionation during abiotic ROS-driven $CH_4$ formation remains to be studied in depth, these results suggest a lower carbon isotope fractionation for ROS-driven $CH_4$ formation than for enzymatic methanogenesis. Together with the observed $CH_4$:$C_2H_6$ ratios, isotopic signatures may therefore serve to differentiate between $CH_4$ formed enzymatically or abiotically on Earth and extraterrestrial planets.

## Discussion

In this work, we demonstrated that the interplay of $Fe^{2+}$ and $H_2O_2$, generated by heat and light, drives $CH_4$ and $C_2H_6$ formation from methylated S-/N-compounds via Fenton chemistry under conditions that were globally prevalent in the Hadean and Archean. As we observed $CH_4$ formation under suboxic and oxic conditions, these mechanisms could, in principle, also contribute to extant $CH_4$ emissions from aqueous environments that were recently shown to correlate with light instead of specific enzymatic pathways[54]. The here described pathways allow $CH_4$ and $C_2H_6$ formation in many aqueous environments including oceans, lakes, rivers, and ponds, delocalized from restricted hotspots for (bio)molecule formation such as hydrothermal vents or ultramafic rocks, in superficial water layers driven by light and throughout the entire water column driven by heat. After the emergence of life, this phenomenon would have greatly intensified in the anoxic Archaean and the subsequent "boring billion"[55,56]. The increasing amounts of biomass provided methylated S-/N-substrates, Fe-chelating biomolecules reducing $Fe^{3+}$ to $Fe^{2+}$ and releasing organic radicals and thus enhance ROS-driven $CH_4$ formation. Possibly, these reactions facilitated elevated $CH_4$ and $C_2H_6$ levels during the Hadean and Archean. These hydrocarbons would have contributed to atmospheric temperatures on Earth and allowed the evolution of life in a liquid hydrosphere which could have influenced the evolution of metabolism by allowing the rise of methanotrophy prior to methanogenesis. This work lays the foundation to explore further the mechanism's role in shaping the evolution of the atmosphere on Earth and other planets and its influence on the current climate change.

## Methods

### General assay conditions

Unless otherwise indicated, 4 mL samples were incubated in closed 20 mL glass vials at 30 °C under a pure nitrogen ($N_2$) atmosphere and subsequently analyzed via gas chromatography (GC).

### Heat assays

In total, 500 mM DMSO and 10 mM $FeSO_4$ were added to 20 mM degassed potassium phosphate buffer (pH 7) in an anaerobic tent. The headspace of the closed vials was then cycled three times with vacuum and $N_2$. Samples were incubated at 37 °C, 57 °C, 77 °C and 97 °C for 6 h

in an incubator in the dark. Optionally, 20 mM citrate, malate, ATP, serine, glucose or pyruvate were also supplemented. $Ca^{2+}$ was added in the form of $CaCl_2$. Samples were measured within the linear range of $CH_4$ formation rate via gas chromatography.

### Light assays

In total, 500 mM DMSO, 2 mM of either $FeCl_3$ or $FeSO_4$ and, optionally, 10 mM citrate were added to 20 mM degassed potassium phosphate buffer (pH 7). Anoxic conditions were generated by drawing vacuum eight times for 1 min and a subsequent filling with $N_2$. For experiments investigating $[Fe(H_2O)_6]^{3+}$ complexes, samples were incubated under anoxic, acidic conditions (20 mM Tris · HCl buffer, pH 3) and supplemented with 500 mM DMSO and either 2 mM $FeCl_3$, 2 mM $FeSO_4$ or 1 mM $FeCl_3$ and 1 mM $FeSO_4$, each. For the investigation of transition metals (Fig. 4a), 2 mM cerium ($CeNH_4SO_4$), manganese ($MnSO_4$), cobalt ($CoNO_3$), nickel ($NiSO_4$), copper ($CuCl_2$) or iron ($FeCl_3$) and 10 mM pH-neutral ascorbate were added to 500 mM DMSO and 20 mM potassium phosphate buffer with an incubation for 1 day. The effect of different wavelengths on $CH_4$ formation (Fig. 4b) was investigated by adding 5 mM $FeCl_3$, 10 mM citrate and 500 mM DMSO to 20 mM potassium phosphate buffer with an incubation for 1 day. For the determination of substrates for $CH_4$ formation (Fig. 4c), 500 mM DMS, methionine, 2-methylthioethanol, Trimethylamine N-oxide or DMSO were added to 10 mM $FeCl_3$ and 100 mM citrate in 20 mM potassium phosphate buffer with an incubation for 3 days. Samples were incubated under air or $N_2$ in the dark or under constant broad-spectrum illumination from light bulbs (Osram, Superlux, Super E SIL 60; $\Phi = 82 \pm 4$ μmol photons m$^{-2}$ s$^{-1}$, $H = 52 \pm 2$ kJ m$^{-2}$ h$^{-1}$; Supplementary Fig. 3) for 1 day. Samples were measured within the linear range of the $CH_4$ formation via gas chromatography. Specific wavelengths were provided by diodes (H2A1 series, Roithner Lasertechnik, Austria) emitting UV-A, blue, cyan, green, red or near-infrared light ($\lambda_{max} = 388$ nm, $\Phi = 35 \pm 1$ μmol photons m$^{-2}$ s$^{-1}$, $H = 36 \pm 2$ kJ m$^{-2}$ h$^{-1}$; $\lambda_{max} = 436$ nm, $\Phi = 45 \pm 1$ μmol photons m$^{-2}$ s$^{-1}$, $H = 45 \pm 1$ kJ m$^{-2}$ h$^{-1}$; $\lambda_{max} = 500$ nm, $\Phi = 64 \pm 4$ μmol photons m$^{-2}$ s$^{-1}$, $H = 55 \pm 3$ kJ m$^{-2}$ h$^{-1}$; $\lambda_{max} = 534$ nm, $\Phi = 63 \pm 1$ μmol photons m$^{-2}$ s$^{-1}$, $H = 50 \pm 1$ kJ m$^{-2}$ h$^{-1}$; $\lambda_{max} = 675$ nm, $\Phi = 45 \pm 4$ μmol photons m$^{-2}$ s$^{-1}$, $H = 29 \pm 3$ kJ m$^{-2}$ h$^{-1}$; or $\lambda_{max} = 868$ nm, $\Phi = 69 \pm 7$ μmol photons m$^{-2}$ s$^{-1}$, $H = 35 \pm 3$ kJ m$^{-2}$ h$^{-1}$) Light intensity was determined using a fiber optic scalar irradiance microsensor[57] connected to a spectrometer (USB4000; Ocean Optics, USA) placed in the center of the incubation vials and calibrated using a spherical light probe (Walz) connected to a LI-250A light meter (Li-Cor Biosciences GmbH, Germany)[58]. Concentration of $Fe^{2+}$ was quantified with the colorimetric ferrozine method[59].

### Bacillus subtilis biomass assays

*B. subtilis* was grown in 500 mL LB media, supplemented with 10 % $H_2O$ or $D_2O$, grown for 36 h at 37 °C and 180 rpm. The obtained culture was collected by three cycles of centrifugation (10 min, 4743 × *g*) and resuspended in 35 mL 20 mM potassium phosphate buffer (pH 7) in order to remove the excess $D_2O$. Biomass was then generated by sonication (4-times, 1 min) and freezing of the samples. Subsequently, 80 mL buffer was supplemented with 10 mL biomass, 20 mM $FeCl_3$ and 50 mM ascorbic acid, saturated with $N_2$ for 30 min and incubated in 100 mL closed glass vials under $N_2$ and constant broad-spectrum illumination for 3 days. The gas headspace was extracted with a syringe and analyzed with regard to $CH_4$ content and $\delta^{2}H$ values.

### Methylocystis hirsuta and Methanothermobacter marburgensis cultivation

*M. hirsuta* growth media contained 0.5 g $Na_2HPO_4 \cdot 2H_2O$, 0.22 g $KH_2PO_4$, 1 g $KNO_3$, 0.4 mg $CaCl_2 \cdot 2H_2O$, 2 mg $MgSO_4 \cdot 7H_2O$ per liter, supplemented with 5 mg $Na_2EDTA$, 0.06 mg $CuCl_2 \cdot 5H_2O$, 2 mg $FeSO_4 \cdot 7H_2O$, 0.1 mg $ZnSO_4 \cdot 7H_2O$, 0.03 mg $MnCl_4 \cdot 4H_2O$, 0.05 mg $H_3BO_3$, 0.2 mg $CoCl_2 \cdot 6H_2O$, 0.02 mg $NiCl_2 \cdot 6H_2O$ and 0.03 mg $Na_2MoO_4 \cdot 2H_2O$

per liter. *M. hirsuta* was cultivated in 100 mL closed glass vials containing 30 mL culture and was incubated at 25 °C and 150 rpm under an air atmosphere. Methane was produced by supplementing 2 L degassed 20 mM potassium phosphate buffer with 1 M DMSO, 25 mM $FeSO_4$ and 50 mM ascorbic acid, incubating the solution under constant illumination in 1 L flasks and collecting the formed $CH_4$ with syringes. *M. hirsuta* cultures were either supplemented with 25 mL light-generated $CH_4$ or 25 mL pure $N_2$. *M. marburgensis* was cultivated as previously described[60].

## Continuous $H_2O_2$ measurements using microsensors

To visualize $H_2O_2$ production in the illuminated anoxic model system, an $H_2O_2$ microsensor was positioned in the solution. The $H_2O_2$ microsensors were built, calibrated and used as described previously[61]. We sealed the vial opening with self-adhesive tape, rigorously bubbled the liquid with $N_2$ and then adjusted a gentle flow of $N_2$ through the headspace to minimize oxygen input from the atmosphere. Light was provided from halogen lamps (KL2500, Schott) at an intensity of 1027 μmol photons $m^{-2}$ $s^{-1}$. We did not attempt to calculate light-dependent $H_2O_2$ production rates due to the open design of the system, which allowed for the exchange of $H_2O_2$ with the headspace across the water interface.

## End-point $H_2O_2$ measurements

After illumination, 290 μL sample was mixed anaerobically with 9 μL Amplex Ultrared (Thermofisher, A36006, 30 μM final concentration) and 1 μL recombinant APEX2 (0.23 μM final concentration). Fluorescence was then measured with a plate reader (BMG ClarioStar™) at 568 nm excitation / 581 nm emission. A calibration curve was established with $H_2O_2$ following the same procedure. To prevent $O_2$-driven $H_2O_2$ generation while sample preparation, all buffers were saturated with $N_2$ and the plate reader was kept at a partial oxygen pressure of 0.1% with an atmospheric control unit (Clariostar, BMG). Before sample preparation, all sample components (20 mM potassium phosphate buffer, DMSO, 1 M citrate and 100 mM $FeCl_3$) were degassed and kept in an anoxic tent overnight.

## Quantification of $CH_4$, $C_2H_6$, $CO_2$, and $H_2$ (GC-FID)

Amounts of formed $CH_4$, $C_2H_6$, $CO_2$ and $H_2$ were determined via headspace analysis using a PerkinElmer® Clarus®690 GC system (GC–FID/TCD) with a custom-made column circuit (ARNL6743). The headspace samples were injected by a TurboMatrixX110 (PerkinElmer Inc, Waltham, USA) autosampler, heating the samples to 45 °C for 15 min prior to injection. The samples were then separated on a HayeSep column (7′ HayeSep N 1/8″ Sf; PerkinElmer®), followed by molecular sieve (9′ Molecular Sieve 13×1/8″ Sf; PerkinElmer®) kept at 60 °C. Subsequently, the gases were detected with a flame ionization detector (FID, at 250 °C) and a thermal conductivity detector (TCD, at 200 °C). The quantification of $CH_4$, $C_2H_6$, $CO_2$ and $H_2$ was based on linear standard curves that were derived from measuring varying amounts of these gases.

## $CH_3OH$ measurements (GC-FID)

$CH_3OH$ was quantified with a GC-FID (Shimadzu GC-2010 Plus, FID-2010 Plus, 280 °C) containing an AOC 20i autosampler and a ZB-WAXplus (Zebron) column (30 m x ∅ = 0.25 mm, df, 0.25 μm). A $H_2O$ sample (1 μL) was injected in the split liner (250 °C, split 5,15,50). The temperature program was kept at 35 °C for 5 min and then increased by 50 °C $min^{-1}$ until 200 °C which was kept for 3 min. Helium served as carrier gas (flow rate: 1.95 ml $min^{-1}$) and the FID was operated with 400 ml $min^{-1}$ synthetic air, 40 ml $min^{-1}$ $H_2$ and 30 ml $min^{-1}$ $N_2$, serving as a makeup gas. For Split 5, a calibration curve ($R^2$ = 0.9931) was generated by diluting $CH_3OH$ (99.9% purity), while an $R^2$ = 0.9981 for split 15 and an $R^2$ = 0.9997 for split 50 was determined.

## δ $^{13}$C stable isotope measurements (GC-C-IRMS)

$δ^{13}C$ values of $CH_4$ were determined by gas chromatography-combustion-isotope ratio mass spectrometry (GC-C-IRMS). Aliquots of headspace gas were transferred to an evacuated sample loop (40 mL) and a cryogenic pre-concentration unit to trap $CH_4$. $CH_4$ was trapped on HayeSep D, separated from interfering compounds by GC and transferred to the GC-C-IRMS. The system consists of a cryogenic pre-concentration unit directly connected to an HP 6890 N GC (He flow rate: 1.8 mL $min^{-1}$; Agilent Technologies, Santa Clara, USA) fitted with a GS-Carbonplot capillary column (30 m * 0.32 mm i.d., $d_f$ 1.5 μm; Agilent Technologies) and a PoraPlot capillary column (25 m * 0.25 mm (i.d.), $d_f$ 8 μm; Varian, Lake Forest, USA). The GC flow was coupled using a press-fit connector to a combustion reactor comprised of an oxidation reactor (ceramic tube ($Al_2O_3$), length 320 mm, inner diameter 0.5 mm, with oxygen-activated Cu/Ni/Pt wires inside; reactor temperature 960 °C) and a GC Combustion III Interface (ThermoQuest Finnigan) to decompose $CH_4$ into $CO_2$. $^{13}C/^{12}C$ ratios were determined with a Delta$^{PLUS}$XL mass spectrometer (ThermoQuest Finnigan, Bremen, Germany). High-purity $CO_2$ (Messer Griesheim, Frankfurt, Germany) was used as the working monitoring gas. $^{13}C/^{12}C$ ratios ($δ^{13}C$ values) are expressed in the conventional δ notation in per mil versus VPDB, calculated as:

$$δ^{13}C_{VPDB} = \left( \frac{\left(\frac{13C}{12C}\right)_{Sample}}{\left(\frac{13C}{12C}\right)_{Standard}} \right) - 1 \qquad (1)$$

$δ^{13}C$ values were corrected using three reference standards of high-purity $CH_4$ with $δ^{13}C$ values of $-54.5 \pm 0.2$ ‰ (Isometric Instruments, Victoria, Canada), $-66.5 \pm 0.2$ ‰ (Isometric Instruments) and $-42.3 \pm 0.2$ ‰ (in-house), calibrated against International Atomic Energy Agency and NIST reference substances.

## δ$^2$H stable isotope measurements (GC-TC-IRMS)

$δ^2H$ values for $CH_4$ were determined using GC-temperature conversion-isotope ratio mass spectrometry (GC-TC-IRMS). The analytical set-up was the same as the one used for $δ^{13}C$ stable isotope measurements except that the He flow rate was changed to 0.6 ml $min^{-1}$ and, instead of combustion to $CO_2$ and $H_2O$, $CH_4$ was thermolytically converted (at 1450 °C) to hydrogen and carbon. After IRMS measurements, the obtained $δ^2H$ values were corrected by using two reference standards of high-purity $CH_4$ with $δ^2H$ values of $-149.9$‰ $\pm$ 0.2‰ (T-iso2, Isometric Instruments) and $-190.6$‰ $\pm$ 0.2‰ (in house). All $δ^2H$ values are expressed in the conventional δ notation in per mil versus Vienna Standard Mean Ocean Water (VSMOW), calculated as

$$δ^2H_{VSMOW} = \left( \frac{\left(\frac{2H}{1H}\right)_{Sample}}{\left(\frac{2H}{1H}\right)_{Standard}} \right) - 1 \qquad (2)$$

## Statistics

Unless indicated otherwise, all experiments were performed with $N$ = 3 replicates (3 biological replicates). To test for significant differences in $CH_4$ formation between two samples, single-factor analysis (two-tailed students $t$ test) of variance (ANOVA) was used.

## Data availability

All data are available in the main text or the supplementary information. The data generated in this study have been deposited on the Edmond database[62], the open repository of the Max Planck Society, under https://doi.org/10.17617/3.6X6JXR.

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

## Acknowledgements

We thank N. Oehlmann, F. Schmidt, H. Addison, A. Lago Maciel, G. Marijan, A. Goldman, K. Guo, F. Arriaza-Gallardo, M. Schneider, C. Hoyer, M. Schroll and M. Greule for practical and theoretical support; I. Bischofs, S. Shima and W. Liesack for providing strains; G. Hochberg, M. Preiner and T. Erb for providing critical comments and feedback. Figures 1A, 2A and 3A, B were created with BioRender.com. This work was supported by German Research Foundation (DFG) grant 446841743 (JGR), SPP2306 (UB, TPD) and KE884 19-1 (FK). J.G.R., L.E and J.M.K are grateful for generous support from the Max Planck Society. L.E thanks the Friedrich Naumann Foundation for support. J.M.K. is grateful for the support from the state of Hesse and the UMR.

## Author contributions
J.G.R. and L.E. conceived the project. J.G.R. supervised and administered the project. J.G.R., F.K., J.M.K., T.D. and L.E. acquired funding. L.E. and J.G.R designed and analyzed the experiments. L.E. performed the experiments. J.M.K. was involved in $H_2O_2$ microsensor and LED experiments (Figs. 3B, 4B, Supplementary Fig. 3). U.B. measured $H_2O_2$ formation (Supplementary Figs. 2, 5 and 10). J.H. measured methanol formation (Supplementary Fig. 6). L.E. and J.G.R. conceptualized, visualized and wrote the original draft. J.G.R., LE, FK and JMK edited the draft. All authors read and reviewed the manuscript.

## Funding

## Competing interests
The authors declare no competing interests.
