## [Peer Review File · Nature Communications]

Methane formation driven by light and heat prior to the origin of life and beyondReviewer #1 (Remarks to the Author):

METHANE FORMATION DRIVEN BY LIGHT AND HEAT PRIOR TO THE ORIGIN OF LIFE

Dear Editor, dear Authors,

Thank you for very much for the opportunity to read your manuscript and hopefully to improve it with my comments.

Overview and general recommendation

Methane is a key molecule for our planet, even more so amid a climate crisis like the one we are facing. Interestingly, this same molecule could have been key in the early stages of the Earth. The formation of methane and other greenhouse gases made it possible to face the so-called young sun paradox, which determined conditions that possibly conditioned the appearance of life on our planet.

The authors present a mechanism for methane formation, via a non-enzymatic pathway, which may help explain the presence of this molecule in less restrictive environments than previously proposed mechanisms.

The authors made a carefully experiment series to test their hypothesis, about the possibility to the abiotic (non-enzymatic) formation of methane under prebiotic conditions. They tested the role of temperature, light, organics addition to the synthesis of methane.

The article is very interesting, but there are some doubts that must be clarified about the text.

A. Originality and significance: if not novel, please include reference
All data presented are novel and represents an original contribution.

B. Data & methodology: validity of approach, quality of data, quality of presentation

The method is adequate, but some details must be further explained. For example, the choice of the DMSO as a methyl donor. Authors mention that it was selected due to its prebiotic formation. However, methionine it is also interesting, since it is an important amino acid that is synthesized in prebiotic experiments. Please, explain why you are using DMSO and not other organic, such as the ones you explored.

The data in the manuscript are very interesting, since methane synthesis is demonstrated under prebiotic conditions. There are some information missing, to make the article clearer. For example, Fig. 1 (D) "[Fe(H₂O)₆]³⁺ photolysis. It is not clear how acidic conditions are obtained. In the case of pH 7, it is clearly described, which is the result of using a phosphate buffer. It is necessary to clarify this point, please.

It could be a problem with my computer, but the figures do not seem to have an adequate resolution, they become blurry when zoomed in to examine the data. If possible, please improve the quality of the graphics.

C. Appropriate use of statistics and treatment of uncertainties

All the experiments were done in triplicate. In some Figures the kind of statistical analysis (two-tailed t-test) is written but it is not mentioned in the Statistics section, please add it.

D. Conclusions: robustness, validity, reliability

Conclusions are based on their experiments. However, I have some little comments.

The photolysis of water effectively produces free radicals, which lately generate hydrogen peroxide. The authors correctly maintain that methane production is promoted by light, and their results support this argument. However, one must be careful with the generalization made in the conclusions (Lines 357-359), since this effect would occur mainly in the superficial layers of water bodies. Well, when light passes through a column of water it is attenuated, part is reflected and another is selectively absorbed, depending on its wavelength.

I suggest being more cautious about the conclusions. For example, instead of saying "potentially influencing the evolution of metabolism", it could be better to say "which could have influenced the evolution of metabolism".

E. Suggested improvements: experiments, data for possible revision

Extended Data Fig. 6. Light-driven formation of CH₄ from DMSO by reduced transition metals. In their experiments authors use various transition metals to assess their effect on methane production, some with variable valences. My question is, why didn't the authors use a key element, like magnesium? Since Mg is abundant in the Earth's crust and plays a fundamental role in the biosphere.

The experiments on the production of methane from biomass are very interesting. However, it seems to me that they are not necessary since the title of the research is limited to the formation of methane before the origin of life. If the authors would like to maintain the experiments, I suggest modifying the title.

F. References: appropriate credit to previous work?

The work is well referenced, and the list includes relevant articles.

G. Clarity and context: lucidity of abstract/summary, appropriateness of abstract, introduction and conclusions

The manuscript is well written and logically organized.

Further Questions

1. Methane is very important as a greenhouse molecule, but its generation is known to be closely related to ethane production. In fact, in your results, you confirm the simultaneous production of both molecules. Although it is true that the half-life of ethane in the atmosphere is very short, compared to that of methane, do you think that in the past the presence of both molecules would have caused a greater green-house effect than estimated? Or the difference in the decomposition rates of the two would be enough to rule out the role for ethane. Specially, when it has been proposed that ethane enabled the warming of the Earth during the Late Archean.

2. The pH estimated in [37] for 4.0 Ga is approximately 6.6 (lower limit 6.2, and upper limit 7.2). While the pH of the cytoplasm is close to 7, is that why you decided to use a buffer with pH 7?

3. Fenton reactions are favoured in environments with acid pHs (pH 3 or less). In the experiments that you carried out, the pH is close to 7 at the beginning of the experiments, and since you are using a buffer solution the variation in pH should be minimal. I wonder if there are any changes in the pH records as the reaction proceeds. If the pH does not acidify, iron oxide should be precipitating, even in small concentrations. Did you observe any precipitation?

4. Methyl donors can be different molecules (i.e. methionine, dimethyl sulfoxide, and trimethylamine). One of the most important molecules, because of its nature (amino acid), is methionine, whose prebiotic synthesis has been demonstrated (References below). In your experiments (these or previous ones) you show a comparison of the effect of some Met-donors in methane yields (Extended Data Fig. 9), but there is no comparison with the DMSO.

- Van Trump, J.E. & Miller S.L. (1972). Prebiotic synthesis of methionine. *Science*, 178 (4063), 859-860.

- Parker, E. T., Cleaves, H. J., Callahan, M. P., Dworkin, J. P., Glavin, D. P., Lazcano, A., & Bada, J. L. (2011). Prebiotic synthesis of methionine and other sulfur-containing organic compounds on the primitive Earth: a contemporary reassessment based on an unpublished 1958 Stanley Miller experiment. *Origins of Life and Evolution of Biospheres*, 41, 201-212.

5. Figure 2. The addition of organic molecules to the studied systems produces different yields of

methane and ethane. Do you have any idea of what is the relationship of the observed differences between methane and ethane?

6. Figure 3. Why is H₂O₂ production shown in arbitrary units? Was it not possible to quantify it?

Minor comments

- All the Figures are essential but need improvement in their quality.
 - Please, check the numbering of the references, specifically 41, it appears as 401.
 - Line 260. You say, "In addition to iron, various other transition metals...", but you don't mention them. Please include the other metals in the text.
 - Line 288. You mention "In short, carboxylic acids like CITRATE.... Please, be careful, citrate is the salt of citric acid.
-
- It might be interesting to have the grow of *Methylocystis hirsuta* in CH₄ different from the one generated in your experiments, just to have a reference.

Reviewer #2 (Remarks to the Author):

I have reviewed the manuscript "Methane formation driven by light and heat prior to the origin of life". The authors build on prior work establishing a novel biological pathway for generating methane, and find that reactive oxygen species and ferrous iron can undergo Fenton reactions with the aid of heat and light.

The methods are thoroughly described. The results are well argued, and are discussed in the context of the rise of methane on the prebiotic Earth. I think this work will be provocative and well received by the community.

Point-by-point reply to the reviewers

Referee 1:

We thank the reviewer for the positive feedback, specifically for highlighting our careful experiments, the relevance of our work and the praise for the writing style and organization. We are equally grateful for the thorough read-through and suggested adjustments.

1.1) Why you are using DMSO and not other organic, such as the ones you explored? However, methionine it is also interesting, since it is an important amino acid that is synthesized in prebiotic experiments.

Reply: We thank the reviewer for this important question. Methionine is a significantly larger and more complex molecule than dimethyl sulfoxide (DMSO). As complex organic molecules necessarily evolved from simpler compounds during chemical evolution¹, it follows that DMSO evolved prior to methionine. Along these lines, several studies in the field of prebiotic chemistry use DMSO as a substrate for the generation of more complex biomolecules²⁻⁴, assuming its early existence during chemical evolution. Moreover, DMSO was already reported to naturally occur at deep-sea vents *in situ*⁵⁻⁷, not only during laboratory-based experiments. Thus, we assume that DMSO is present at relevant amounts prior to methionine during the evolution of Earth. However, the reviewer rightfully mentions the importance of methionine, therefore we have moved the data regarding the different tested substrates into the main text (see Fig. 4) that shows CH₄ formation from methionine.

1.2) Fig. 1 (D) "[Fe(H₂O)₆]³⁺ photolysis. How were acidic conditions obtained?

Reply: Acidic conditions were obtained by using a 20 mM TRIS · HCl buffer. We added this information into the appropriate Methods section:

"For experiments investigating [Fe(H₂O)₆]³⁺ complexes, samples were incubated under anoxic, acidic conditions (20 mM Tris · HCl buffer, pH 3) and supplemented with 500 mM DMSO"

1.3) The figures do not seem to have an adequate resolution, they become blurry when zoomed in to examine the data. Please improve the quality of the graphics.

Reply: We thank the reviewer for pointing this out. High-resolution figures are uploaded with the revised manuscript as separate files.

1.4) In some Figures the kind of statistical analysis (two-tailed t-test) is written but it is not mentioned in the Statistics section, please add it.

Reply: We thank the reviewer for this remark. This information is now added to the Statistics section:

“To test for significant differences in CH₄ formation between two samples, single-factor analysis (two-tailed students t-test) of variance (ANOVA) was used.”

1.5) One must be careful with the generalization made in the conclusions (Lines 357-359), since this effect would occur mainly in the superficial layers of water bodies. Well, when light passes through a column of water it is attenuated, part is reflected and another is selectively absorbed, depending on its wavelength.

Reply: We apologize for our imprecise statement. We agree with the reviewer that the light-driven CH₄ formation pathways would mainly be limited to superficial water layers. In contrast, the heat-driven CH₄ formation that we describe does not depend on illumination and hence can occur throughout the entire water column. To avoid confusion, we modified the corresponding sentence of the Discussion section:

“The here described pathways allow CH₄ and C₂H₆ formation in many aqueous environments including oceans, lakes, rivers, and ponds, delocalized from restricted hotspots for (bio)molecule formation such as hydrothermal vents or ultramafic rocks, in superficial water layers driven by light and throughout the entire water column driven by heat.”

1.6) I suggest being more cautious about the conclusions. For example, instead of saying “potentially influencing the evolution of metabolism”, it could be better to say “which could have influenced the evolution of metabolism”.

Reply: We thank the reviewer for this suggestion. To present more cautious conclusions, we have changed it to:

“These hydrocarbons would have contributed to atmospheric temperatures on Earth and allowed the evolution of life in a liquid hydrosphere which could have influenced the evolution of metabolism by allowing the rise of methanotrophy prior to methanogenesis.”

1.7) Extended Data Fig. 6. Light-driven formation of CH₄ from DMSO by reduced transition metals. My question is, why didn't the authors use a key element, like magnesium? Since Mg is abundant in the Earth's crust and plays a fundamental role in the biosphere.

Reply: We thank the reviewer for this valuable suggestion, as magnesium (Mg²⁺) is indeed a very abundant environmental element. Therefore, we investigated the role of Mg²⁺ in an additional experiment we would like to add to the manuscript (Supplementary Fig. 8): Here, we find that Mg²⁺ does not contribute to light-driven CH₄ formation. In contrast, it acts in a similar way as Ca²⁺ (Fig. 2C), as Mg²⁺ also can be chelated by citrate, thereby outcompeting Fe²⁺ chelation and consequently reducing obtained CH₄ amounts. We added the following part to the manuscript:

“Most importantly, these data highlight that a light- and ROS-driven iron cycle can facilitate high rates of CH₄ formation, even in the presence of O₂ and the absence of detectable Fe²⁺, which opens the possibility of widespread abiotic CH₄ production after the great oxidation event as well as in diverse modern habitats. Next, we investigated the role of the alkali metal magnesium (Mg²⁺) due to its high environmental abundance

and found that Mg^{2+} does not facilitate CH_4 formation in illuminated buffer containing DMSO and citrate (Supplementary Fig. 8). Upon additional Fe^{3+} supplementation, Mg^{2+} also decreased CH_4 formation rates by replacing Fenton-promoting Fe^{3+} -citrate complexes by Mg^{2+} -citrate complexes, thereby acting similar to Ca^{2+} that was demonstrated to decrease heat-driven CH_4 formation (Fig. 2C)."

1.8) The experiments on the production of methane from biomass are very interesting. However, it seems to me that they are not necessary since the title of the research is limited to the formation of methane before the origin of life. If the authors would like to maintain the experiments, I suggest modifying the title.

Reply: We thank the reviewer for this very helpful advice. We have changed the title to:

"Methane formation driven by light and heat prior to the origin of life **and beyond**"

1.9) Methane is very important as a greenhouse molecule, but its generation is known to be closely related to ethane production. In fact, in your results, you confirm the simultaneous production of both molecules. Although it is true that the half-life of ethane in the atmosphere is very short, compared to that of methane, do you think that in the past the presence of both molecules would have caused a greater green-house effect than estimated? Or the difference in the decomposition rates of the two would be enough to rule out the role for ethane. Specially, when it has been proposed that ethane enabled the warming of the Earth during the Late Archean.

Reply: We thank the reviewer for this question. Indeed, we believe that ethane would also contribute to warming of the earth. We modified the introduction to:

"Atmospherically accumulated CH_4 **and ethane (C_2H_6)** might also explain the "faint young sun paradox", which describes the apparent contradiction of a fainter sun (70% – 83% of the current solar energy output) but a climate that was at least as warm as today during early Earth (4.5–2.5 Ga ago)^{2–4}."

Furthermore, the contribution of ethane is mentioned in the last paragraph of the section "Methane is formed under abiotic conditions":

"Besides CH_4 , C_2H_6 is considered an important factor in keeping the early Earth warm, since C_2H_6 absorbs from 11 to 13 μm in an atmospheric window (roughly 8-13 μm) where H_2O and CO_2 do not absorb strongly². Together, the hydrocarbons produced by these pathways might offer a solution to the "faint young sun paradox"^{3,4}"

1.10) The pH estimated in [37] for 4.0 Ga is approximately 6.6 (lower limit 6.2, and upper limit 7.2). While the pH of the cytoplasm is close to 7, is that why you decided to use a buffer with pH 7?

Reply. The reviewer is correct. A pH of 7 falls within the estimated pH range of Hadean and Archean oceans. As Fenton chemistry is favored under acidic conditions, we decided to use a pH value at the upper limit of the estimated pH range avoid an overestimation of the observed CH_4 formation rates. Furthermore, a pH of 7 serves as

a standard for proof-of-principle *in vitro* studies, while subsequent field studies should account for the precise environmental and local pH values. Along these lines, a neutral pH also allows for a better comparison between our previous publication describing CH₄ formation in living organisms (thereby in the cytoplasm)⁸ and this project.

1.11) Fenton reactions are favoured in environments with acid pHs (pH 3 or less). In the experiments that you carried out, the pH is close to 7 at the beginning of the experiments, and since you are using a buffer solution the variation in pH should be minimal. I wonder if there are any changes in the pH records as the reaction proceeds. If the pH does not acidify, iron oxide should be precipitating, even in small concentrations. Did you observe any precipitation?

Reply: The reviewer is right for pointing out that iron oxide should be precipitating under a neutral pH. Indeed, we observed precipitation under pH-neutral conditions, indicating low levels of formed Fe(OH)₂. Upon additional citrate supplementation, precipitation was prevented due to the iron-chelating effect of citrate and other Fe-chelators. We did not observe any significant pH changes over time. To clarify this point, we added the following sentences to the manuscript:

“To investigate CH₄ formation under abiotic conditions (Fig. 1A), we designed a chemical model system consisting of a nitrogen atmosphere, a potassium phosphate-buffered solution (pH 7, expected during the Archean at 4.0 Ga³⁷) supplemented with Fe²⁺ and the abiotically formed DMSO which serves as methyl donor for ROS-driven CH₄ formation. Over the course of the experiments, no pH change was observed, while small amounts of Fe(OH)₂ precipitated.”

...

“The Ca²⁺ concentration-dependent decrease of the heat-driven CH₄ formation rate supports the role of citrate as a Fenton-promoting Fe²⁺-chelator, which was further indicated by citrate dissolving any ferruginous precipitate.”

1.12) Methyl donors can be different molecules (i.e. methionine, dimethyl sulfoxide, and trimethylamine). One of the most important molecules, because of its nature (amino acid), is methionine, whose prebiotic synthesis has been demonstrated. In your experiments (these or previous ones) you show a comparison of the effect of some Met-donors in methane yields (Extended Data Fig. 9), but there is no comparison with the DMSO.

Reply: We thank the reviewer for this very important comment. We have added DMSO as a comparison to Fig. 4C.

1.13) Figure 2. The addition of organic molecules to the studied systems produces different yields of methane and ethane. Do you have any idea of what is the relationship of the observed differences between methane and ethane?

Reply: We thank the reviewer for this question. In theory, the CH₄:C₂H₆ ratio should remain constant if only accounting for methyl radicals forming either CH₄ or C₂H₆. Although we do not have clear evidence for this phenomenon, we hypothesize that the heat-generated hydroxyl/methyl radicals can facilitate the decomposition of the added

substrates, directly releasing C₂H₆. Depending on the structure of the substrate, C₂H₆ may be formed in a similar way as the organoradical-driven CH₄ formation depicted in Fig. 3 – directly from the substrate, beyond the C₂H₆ formation from two methyl radicals. These mechanistic insights are currently further investigated in the lab and are beyond the scope of this study.

1.14) Figure 3. Why is H₂O₂ production shown in arbitrary units? Was it not possible to quantify it?

Reply: For continuous H₂O₂ measurements, we did not attempt to calculate light-dependent H₂O₂ production rates due to the open design of the system and continuous bubbling with N₂, which allows for the exchange of H₂O₂ with the headspace across the water interface. We have highlighted this in the Methods section:

“We did not attempt to calculate light-dependent H₂O₂ production rates due to the open design of the system, which allowed for the exchange of H₂O₂ with the headspace across the water interface.”

However, we quantified produced H₂O₂ levels via end-point measurements in anoxic and closed samples (Supplementary Figs. 2, 5).

Minor comments

1.15) All the Figures are essential but need improvement in their quality.

Reply: We thank the reviewer for pointing this out. Individual high-resolution figures will be uploaded with the revised manuscript.

1.16) Please, check the numbering of the references, specifically 41, it appears as 401.

Reply: We thank the reviewer for identifying this typing error. We removed this mistake and again checked the reference numbering.

1.17) Line 260. You say, “In addition to iron, various other transition metals...”, but you don't mention them. Please include the other metals in the text.

Reply: All other metals are now mentioned:

“Besides iron, the transition metals copper, cerium, cobalt, nickel and manganese were reported to drive Fenton chemistry^{45,46}”

1.18) Line 288. You mention “In short, carboxylic acids like CITRATE.... Please, be careful, citrate is the salt of citric acid.

Reply: The reviewer is right for pointing this out. We exchanged “carboxylic acids” with “carboxylates” (changes highlighted in yellow in the revised manuscript).

1.19) It might be interesting to have the growth of *Methylocystis hirsuta* in CH₄ different from the one generated in your experiments, just to have a reference.

Reply: Prior to this experiment, we cultivated *M. hirsuta* by adding 20 % pure CH₄ to the headspace. Growth behavior and final OD₆₀₀ did not change in comparison to the CH₄ generated in our experiments.

Referee 2:

I have reviewed the manuscript "Methane formation driven by light and heat prior to the origin of life". The authors build on prior work establishing a novel biological pathway for generating methane, and find that reactive oxygen species and ferrous iron can undergo Fenton reactions with the aid of heat and light.

The methods are thoroughly described. The results are well argued, and are discussed in the context of the rise of methane on the prebiotic Earth. I think this work will be provocative and well received by the community.

Reply: We thank the reviewer for the positive assessment, highlighting and the well supported arguments as well as the relevance of our work for the scientific community.

References:

1. Dickerson, Richard E. "Chemical evolution and the origin of life." *Scientific American* 239.3 (1978): 70-87.
2. Hermes-Lima, Marcelo, et al. "Pyrophosphate and Adenosine 5'-Diphosphate Synthesis from Phospho (enol) pyruvate: Catalysis by Phosphate Minerals and Modulation by Dimethyl Sulfoxide." *Journal of molecular evolution* 44 (1997): 106-111.
3. Lv, Yunhe, et al. "Copper-catalyzed annulation of amidines for quinazoline synthesis." *Chemical Communications* 49.57 (2013): 6439-6441.
4. Ma, Jin-Tian, et al. "Access to 2-arylquinazolines via catabolism/reconstruction of amino acids with the insertion of dimethyl sulfoxide." *Chemical Communications* 57.44 (2021): 5414-5417.
5. Rogers, Karyn L., and Mitchell D. Schulte. "Organic sulfur metabolisms in hydrothermal environments." *Geobiology* 10.4 (2012): 320-332.
6. Beinart, Roxanne A., et al. "The bacterial symbionts of closely related hydrothermal vent snails with distinct geochemical habitats show broad similarity in chemoautotrophic gene content." *Frontiers in Microbiology* 10 (2019): 1818.
7. Zeng, Xiang, Karine Alain, and Zongze Shao. "Microorganisms from deep-sea hydrothermal vents." *Marine Life Science & Technology* 3 (2021): 204-230.
8. Ernst, Leonard, et al. "Methane formation driven by reactive oxygen species across all living organisms." *Nature* 603.7901 (2022): 482-487.